# Whole-Genome DNA Methylation Profiling of CD14+ Monocytes Reveals Disease Status and Activity Differences in Crohn’s Disease Patients

**DOI:** 10.3390/jcm9041055

**Published:** 2020-04-08

**Authors:** Andrew Y.F. Li Yim, Nicolette W. Duijvis, Mohammed Ghiboub, Catriona Sharp, Enrico Ferrero, Marcel M.A.M. Mannens, Geert R. D’Haens, Wouter J. de Jonge, Anje A. te Velde, Peter Henneman

**Affiliations:** 1Department of Clinical Genetics, Amsterdam University Medical Centers, University of Amsterdam, Genome Diagnostics Laboratory, Amsterdam Reproduction & Development, 1105 AZ Amsterdam, The Netherlands; 2R&D GlaxoSmithKline, Stevenage SG1 2NY, UK; m.ghiboub@amsterdamumc.nl (M.G.); catriona.h.sharp@gsk.com (C.S.); enrico.ferrero@novartis.com (E.F.); 3Tytgat Institute for Liver and Intestinal Research, Amsterdam University Medical Centers, University of Amsterdam, Amsterdam Gastroenterology & Metabolism, 1105 BK Amsterdam, The Netherlands; n.duijvis@gmail.com (N.W.D.); w.j.dejonge@amsterdamumc.nl (W.J.d.J.); 4Department of Gastroenterology and Hepatology, Amsterdam University Medical Centers, University of Amsterdam, Amsterdam Gastroenterology & Metabolism, 1105 AZ Amsterdam, The Netherlands; g.dhaens@amsterdamumc.nl; 5Department of Surgery, University Clinic of Bonn, 53127 Bonn, Germany

**Keywords:** DNA methylation, CD14+ monocytes, Crohn’s disease, CD-activity

## Abstract

Crohn’s disease (CD) is a multifactorial incurable chronic disorder. Current medical treatment seeks to induce and maintain a state of remission. During episodes of inflammation, monocytes infiltrate the inflamed mucosa whereupon they differentiate into macrophages with a pro-inflammatory phenotype. Here, we sought to characterize the circulating monocytes by profiling their DNA methylome and relate it to the level of CD activity. We gathered an all-female age-matched cohort of 16 CD patients and 7 non-CD volunteers. CD patients were further subdivided into 8 CD patients with active disease (CD-active) and 8 CD patients in remission (CD-remissive) as determined by the physician global assessment. We identified 15 and 12 differentially methylated genes (DMGs) when comparing CD with non-CD and CD-active with CD-remissive, respectively. Differential methylation was predominantly found in the promoter regions of inflammatory genes. Comparing our observations with gene expression data on classical (CD14^++^CD16^-^), non-classical (CD14^+^CD16^++^) and intermediate (CD14^++^CD16^+^) monocytes indicated that while 7 DMGs were differentially expressed across the 3 subsets, the remaining DMGs could not immediately be associated with differences in known populations. We conclude that CD activity is associated with differences in DNA methylation at the promoter region of inflammation-associated genes.

## 1. Introduction

Crohn’s disease (CD) is a debilitating disorder belonging to the family of inflammatory bowel disease (IBD). CD is characterized by episodes of transmural inflammation that can affect any part of the entire gastrointestinal tract. Inflammatory episodes typically manifest as a disproportionate immune response against the commensal microbiota [1], which is accompanied by infiltration of leukocytes into the inflamed intestinal mucosa [2]. Despite the extensive research performed on CD, it remains an incurable disease whose etiology and pathogenesis is not fully understood. Treatment regimens therefore aim to reduce inflammation by inducing and subsequently maintaining a state of remission.

Genome-wide association studies (GWAS) have made it clear that genetics alone does not fully explain heritability in CD [3,4,5]. As such, CD has been classified as a complex disorder whose etiology is likely to be a combination of genetic [4], epigenetic [6,7] and other environmental factors. Epigenetics pertain to mitotically heritable changes that affect the readability of the genome that are not caused by changes to the genetic sequence. DNA methylation is one of the most studied epigenetic marks and represents the presence of a methyl group on a cytosine [8]. Functionally, the presence of DNA methylation in the promoter area is often inversely correlated with gene expression [9,10,11], which in certain cases was found to be a causal relationship [12,13]. Previous epigenetic studies reported differences in the DNA methylome of peripheral blood or peripheral blood mononuclear cells (PBMCs), with differentially methylated loci occurring in genes associated with inflammatory pathways [14,15,16]. Here, we sought to build on the previous studies by focusing on an individual immune cell type: monocytes.

Monocytes can differentiate into macrophages or dendritic cells (DCs), which altogether are known as the mononuclear phagocyte system (MPS) [17]. Blood monocytes are typically identified by their cell-surface expression of CD14, a pattern recognition receptor that acts as a co-receptor for detecting bacterial lipopolysaccharides [18]. Further sub-classification based on the expression of CD16, a type III Fcγ receptor, led to the definition of classical (CD14^++^CD16^-^), non-classical (CD14^+^CD16^++^) and intermediate (CD14^++^CD16^+^) monocytes [19,20,21]. Where classical monocytes were typified by their phagocytic behavior and innate immune response, intermediate monocytes were found to be involved in cytokine secretion, antigen presentation and apoptosis, while non-classical monocytes were associated with adhesion, complement and Fc gamma-mediated phagocytosis [22,23,24].

Circulating monocytes alongside the intestinal macrophages and DCs have been implicated in the pathogenesis of IBD [25,26,27,28,29,30,31], with a recent study indicating that 170 CD-associated loci obtained from GWAS coincide with the gene co-expression networks from monocytes [32]. Relative to non-CD individuals or CD patients in remission, blood monocytes obtained from CD patients with active disease were more prone to secrete the inflammatory cytokines IL6 [33], CCL2 [34], and IL1β [34]. Subsequent flow cytometry studies identified fewer non-classical monocytes, but increased classical and intermediate monocytes among CD patients relative to healthy individuals [28,31]. The same held true when comparing CD patients with active disease (CDAI > 150) versus CD patients with quiescent disease (CDAI < 150) [28,31]. It has been suggested that the classical monocytes infiltrate the mucosa during inflammatory episodes of IBD [35] whereupon they differentiate into macrophages that display an inflammatory phenotype [36]. Among IBD patients, such an increased presence of inflammatory macrophages has been observed in the gut, which was more prominent in patients with active CD [27,28,37].

In this study, we characterized the DNA methylome of CD14^+^ monocytes in CD patients. We identified differences in methylation between female CD patients and non-CD volunteers as well as between active and remissive CD patients, and associated them with differences in cellular composition observed in monocytes.

## 2. Experimental Section

### 2.1. CD14 Cells Isolation

The assembly of this cohort was approved by the medical ethics committee of the Academic Medical Hospital (2014_178#C20142104, dated 7 November 2014). Written informed consent was obtained from both the CD patients and control subjects.

Peripheral blood was collected in heparin tubes (BD Vacutainer, Plymouth, United Kingdom) after which peripheral blood mononuclear cells (PBMCs) were obtained by density gradient centrifugation using Ficoll (Invitrogen, Thermo Fisher Scientific, Grand Island, New York, United States of America). CD14^+^ cells were positively selected from PBMCs using CD14 Microbeads according to the manufacturer’s instructions (Miltenyi Biotec, Leiden, The Netherlands). Resulting PBMCs were then stored in PBS (Fresenius Kabi, Graz, Austria) at −80 °C until the cohort was fully assembled.

### 2.2. DNA Isolation and Methylation Analysis

Genomic DNA was extracted using the QIAamp DNA Mini Kit (Qiagen, Venlo, The Netherlands) according to the manufacturer’s instructions and stored at 4 °C. Subsequent bisulfite conversion of the DNA was performed using the Zymo EZ DNA Methylation™ kit (Zymo Research, Irvine, California, United States of America) according to the manufacturer’s protocol prior to hybridization onto the Illumina HumanMethylation 450k BeadChip array for whole-genome DNA methylation profiling.

Raw methylation data was imported into the R statistical programming environment (v3.6.2) [38] using the Bioconductor (v3.10) package minfi (v1.30) [39] after which quality control was performed using MethylAid (v1.18) [40] and shinyMethyl (v1.20) [41]. For statistical analyses, M-values were used (1), whereas for visualization purposes β-values (percentage methylation) were used (2).
(1)$$M=\log_{2}\left(\frac{\max\left({\mathrm{methylated}}_{i}\right)+1}{\max\left({\mathrm{unmethylated}}_{i}\right)+1}\right),$$
(2)$$\beta =\frac{\max\left({\mathrm{methylated}}_{i},0\right)\;}{\max\left({\mathrm{unmethylated}}_{i},0\right)+\max\left({\mathrm{methylated}}_{i},0\right)},$$
where methylated_i_ and unmethylated_i_ represent the signal intensity obtained from the green and red channel, respectively, as described in [42].

Differential methylation analyses were performed using limma (v3.36) [43] and DMRcate (v1.16) [44] to identify differentially methylated probes (DMPs) and regions (DMRs), respectively. DMPs were defined as probes with a Benjamini–Hochberg (BH)-adjusted *p*-value < 0.05. DMRs were defined as regions with a Stouffer statistic < 0.05. Probes were annotated using the annotation file provided by Illumina (v1.2). We constructed two separate linear models where we compared CD with non-CD and the CD-active with CD-remissive while correcting for age (3).
(3)$$\sim {\mathrm{CD}}_{\mathrm{status}}+\mathrm{age}$$

Comparisons included CD patients against non-CD controls, and CD patients with active disease against CD patients in remission. Reported chromosomal coordinates were based on the genome build GRCh37. Differentially methylated genes (DMGs) were identified by aggregating *p*-values of the individual probes associated per gene using Brown’s method [45] as implemented in EmpiricalBrownsMethod (v1.14.0) [46] and identifying the genes with a BH-adjusted *p*-value < 0.05. Briefly, Brown’s method aggregates *p*-values and is therefore used frequently in meta-analyses [45]. Unlike the related Fisher’s combined probability test, which assumes independence between the individual tests, Brown’s method accounts for the dependence between the individual tests [47]. Given the correlated nature of CpGs within close proximity [48], Brown’s method was deemed more suitable than Fisher’s method. Visualizations were generated using the ggplot (v3.2.1) [49] and ggbio (v1.32) [50] packages with gene features obtained from TxDb.Hsapiens.UCSC.hg19.knownGene (v3.2.2) [51] and CpG island features obtained from AnnotationHub (v2.18.0) [52], both of which were sourced from the University of California, Santa Cruz (UCSC) Genome Browser [53].

### 2.3. Monocyte Gene Expression Data

Gene expression data was obtained from the Gene Expression Omnibus [54] dataset GSE107011 [55], which contained a paired-ended RNA-sequencing data from different cell types isolated from peripheral blood from two male and two female healthy individuals. We downloaded the raw reads on the classical (CD14^++^CD16^-^), non-classical (CD14^+^CD16^++^), and intermediate (CD14^++^CD16^+^) monocytes from the Sequence Read Archive (SRA) [56] and aligned them against the human genome (GRCh37) using the STAR short read mapper (v2.7.1a) [57]. Subsequent post-processing was done using SAMtools (v1.9) after which reads mapped per gene were counted using featureCounts (v1.6.4) from the Subread package [58,59]. Raw counts were imported into the R statistical programming environment after which normalization and statistical analysis was performed using DESeq2 (v1.24) [60]. To test for difference across monocyte subsets, we therefore utilized a likelihood ratio test as implemented in DESeq2 where we defined the full model and the reduced model as (4) and (5):(4)~individual+monocyte,
(5)~individual,
where individual represents the donor and monocyte subset. Subsequent comparative analyses were done using the default Wald test as implemented in DESeq2 where we compared classical with non-classical, classical with intermediate and intermediate with non-classical monocytes.

## 3. Results

### 3.1. CD-Associated Differential Methylation

A cohort of 23 female individuals were assembled, consisting of 16 CD patients and 7 non-CD healthy volunteers. CD patients were selected to include 8 active and 8 remissive CD patients that visited the outpatient clinic at the IBD department in Amsterdam UMC, The Netherlands. Active CD was determined by a physician global assessment [61], where the assessment was based on clinical, such as the Harvey Bradshaw Index (HBI), endoscopic or magnetic resonance imaging (MRI), and biochemical parameters, such as C-reactive protein (CRP; median CRP > 12) and/or fecal calprotectin (Table 1).

### 3.2. CD-Associated Differential Methylation

We first compared the CD with non-CD samples but found no probes that passed the threshold for statistical significance (Appendix A). Notably, the 50 most differentially methylated probes revealed visual, albeit minor, differences between CD and non-CD patients (Figure 1a). Systematically searching for differentially methylated regions (DMRs) yielded no statistically significant DMRs either. However, visualizing the DMR with the lowest Stouffer statistic (chr7:51,470,953-51,471,981; Stouffer-statistic = 0.50) displayed continuous hypermethylation among the CD samples relative to the non-CD samples for 8 CpGs (Appendix A). Trying to annotate this DMR to a particular gene proved inconclusive due to its large distance (> 100 kb) to the nearest gene, Cordon-Blue WH2 Repeat Protein (*COBL*).

We searched for genes that were enriched for CpGs with low *p*-values. To that end, we annotated the CpGs to their respective genes and aggregated the *p*-values by means of the Brown’s method [45]. This approach yielded 15 statistically significant differentially methylated genes (DMGs) (Figure 1b). Visualization of the difference in methylation suggested visually consistent, yet minor, differences in methylation (Figure 1c). *MPIG6B*, *GSTT1*, *SLFN13*, *SPI1*, *ZNF572*, *LOC150381*, and *G0S2* displayed hypomethylation in the region surrounding the transcription start site (TSS), which we considered the promoter region, whereas *ZADH2*, *DRD4*, *MPEG1*, and *SLC26A4* displayed hypomethylation within the gene body. Conversely, *PDCD1* and *MPEG1* displayed promoter hypermethylation with *SLC17A9* and *LOC286002* displaying hypermethylation within the gene body. Notably, *MPIG6B, GSTT1, ZADH2, DRD4, SLFN13, SLC17A9, SLC26A4, SPI1,* and *LOC150381* displayed the largest difference in methylation within a densely populated region of CpGs, which the UCSC annotated as a CpG island.

### 3.3. Differential Methylation Associated with Disease Activity in CD Monocytes

As we had more granular information on CD activity, we investigated the intra-CD differences by comparing CD patients with active disease against CD patients in remission (Appendix A). Like the previous comparisons, none of the individual probes or continuous regions of probes were statistically significant after correcting for multiple testing. However, visualizing the top 50 most differentially methylated probes suggested again visible but minor differences (Figure 2a). Utilizing the Brown’s method for aggregating *p*-values, we identified 12 DMGs that were significantly associated with CD activity (Figure 2b). Hypomethylation was observed for *NNAT*, *TRIP6*, and *LOC387647* in the promoter and for *HCP5* in the gene body (Figure 2c). By contrast, hypermethylation was observed for *MPIG6B*, *KRT3CAP*, *FAM24B*, *ZNF153* and *PRAP1* in the promoter (Figure 2c). For *NNAT*, *MPIG6B*, *KRTCAP3*, *TRIP6, LOC387647*, *SSTR4*, *FAM24B*, and *ZNF154* the largest differences in methylation were found in regions annotated as CpG islands.

While all CD-remissive samples were obtained from patients on some kind of medication (anti-TNF, corticosteroid, thiopurine, mercaptopurine, celecoxib, or questran), two CD-active samples were obtained from patients that were not on any medical treatment at the time of sampling. We therefore investigated whether a medication effect was observable for the aforementioned DMGs by means of principal component analysis. We observed no separate clustering of the samples on medication relative to the other samples, suggesting that any effect of the medication did not manifest visibly in the methylome of the DMGs (Figure 2d).

Taken together, we have identified in total 26 genes that were differentially methylated between CD and non-CD or between CD-active and CD-remissive (Table 2). When comparing the DMGs from the CD with non-CD comparison with the DMGs obtained from the active with remissive comparison, we identified one gene that was present in both comparisons, namely *MPIG6B* (Figure 3a). Somewhat surprisingly, visualizing the methylation pattern of *MPIG6B* for all three groups, indicated that CD patients with active disease displayed a methylome more similar to non-CD patients as compared to CD patients in remission (Figure 3b).

### 3.4. Differences in Methylation may be Associated with Disease Dynamics in Monocyte Populations

From previous studies we know that CD patients compared with non-CD individuals, as well as CD patients with active disease compared with CD patients in remission, present an increased classical and intermediate monocyte population and a reduced non-classical monocyte population in peripheral blood [28,31]. We therefore sought to identify which DMGs were potentially due to differences in monocyte populations. To investigate this, we analyzed the expression of the DMGs for all the three monocyte subsets using an external RNA-sequencing dataset (GSE107011 [55]).

Monocyte gene expression data was available for 9 CD-associated DMGs, namely *MPEG1*, *G0S2*, *ZNF572*, *ZADH2*, *SLFN13*, *PDCD1*, *SPI1*, *SLC17A9*, and *MS4A3*, and 7 CD-activity associated DMGs, namely *SERPINF1*, *HCP5*, *TRIOBP*, *KRTCAP3*, *ZNF154*, *TRIP6*, and *FAM24B*. By performing a likelihood ratio test, we identified that the CD-associated DMGs *MPEG1*, *G0S2*, *ZNF572* and *ZADH2* (Figure 4a) and the CD-activity associated DMGs *SERPINF1* and *HCP5* (Figure 4b) were significantly differentially expressed among the monocyte populations. Classical monocytes were characterized by high *MPEG1* and *ZNF572* expression, intermediate monocytes were characterized by high *ZADH2* expression, and non-classical monocytes were characterized by low *G0S2* and *HCP5*. Notably, all three subsets expressed *SERPINF1* in a different fashion. By contrast, CD-associated DMGs *SLFN13*, *PDCD1*, *SPI1*, *SLC17A9*, and *MS4A3* and CD-activity associated DMGs *TRIOBP*, *KRTCAP3*, *ZNF154*, *TRIP6*, and *FAM24B* were not significantly differentially expressed.

## 4. Discussion

In this study, we investigated the DNA methylome of CD14^+^ monocytes and its relation to CD activity. To that end, we performed two analyses. First, we compared CD14^+^ monocytes from CD patients with non-CD volunteers and second, we compared CD patients with active disease against those in remission. At a genome-wide level, we identified no statistically significant DMPs and DMRs for both comparisons, suggesting minor differences methylation across the three groups. Despite the lack of genome-wide statistical significance, our search for genes that were enriched for low nominal *p*-values yielded 15 and 12 genes for the CD vs. non-CD and CD-active vs. CD-remissive comparisons, respectively. Notably, most of the CD-associated (9 out of 15) and the CD-activity associated (8 out of 12) DMGs presented the largest differences in CpG islands. Cross-referencing our observations with differences in gene expression among monocyte subpopulations suggested that while 4 out of 9 CD-associated and 2 out of 7 CD-activity associated were potentially associated to changes in the underlying monocyte populations, 5 CD-associated and 5 CD-activity were not.

Several DMGs have been associated with CD or ulcerative colitis (UC) or phenotypes thereof. We reported previously that *SERPINF1* was differentially methylated and expressed when comparing ileal fibroblasts obtained from stenotic tissue with non-inflamed tissue from CD patients [62]. Similarly, *PRAP1* was found to be hypermethylated and downregulated in mucosal biopsies obtained from treatment naïve UC patients relative to control patients [63]. At the level of genomics, a meta-analysis suggested that the *GSTT1* null mutation was significantly associated with susceptibility to IBD [64]. Unaffected ileal samples obtained from carriers of the *NOD2* CD-risk allele displayed increased gene expression of *DRD4* [65], whereas *Nod2* double knockout mouse macrophages displayed a higher *Ms4a3* expression relative to wildtype after lipopolysaccharide treatment [66]. Transcription-wise, *G0S2* gene expression in mucosal biopsies was found to be predictive of clinical response to infliximab [67].

Functionally, the DMGs were not found to be overrepresented for gene sets using the STRING database [68], indicating that the DMGs do not represent clear functional modules or cellular pathways. Nonetheless, CD-associated DMGs *PDCD1*, *SPI1*, *SLC26A4, MPEG1* and *MPIG6B* as well as CD-activity associated DMGs *TRIP6*, *SSTR4*, HCP5, and *SLC17A9* have been implicated in immunological functions. The PDCD1 protein is involved in the programmed cell death pathway [69], whose inhibition benefits sepsis-associated microbial clearing in murine macrophages [70,71]. SPI1 (also known as PU.1) is a known regulator of myeloid and B-lymphoid cell development [72] but has also been described as pro-inflammatory as it is capable of upregulating the cytokine IL6 in the presence of lipopolysaccharides (LPS) [73]. *SLC26A4* encodes pendrin, an anion exchange protein whose clinical relevance is mostly described within the context of hearing impairment [74]. Nonetheless, whole genome bisulfite sequencing and RNA-sequencing analysis of mucosal biopsies of UC patients with non-UC patients indicated promoter hypomethylation and upregulated expression [63], which is in agreement with the observations made in this study. *MPEG1* encodes Perforin-2, which is a protein expressed in phagocytes involved in the innate immune response by forming pores in bacteria [75,76]. *MPIG6B* expression in platelets has been associated with a decreased aggregative capability in vitro [77]. Platelet count is typically positively correlated with CD activity [78] or colonic inflammation [79]. Notably, our results show hypomethylation of a CpG island in the *MPIG6B* promoter when comparing CD with non-CD, yet hypermethylation when comparing CD-active with CD-remissive. This observation would require further mechanistic studies to investigate the role of *MPIG6B* methylation on the inflammatory phenotype in monocytes. *TRIP6* encodes a member of the RIP kinase family involved in inflammation through the NOD-like receptor signaling [80]. NOD-like receptors remain an interesting target for auto-inflammatory diseases due to their role in the assembly of the inflammasome [81] and autophagy [82]. Similarly *SSTR4* has been implicated in inflammation and nociception in the gastrointestinal tract [83]. *SLC17A9* encodes a vesicular nucleotide transporter whose primary function is the export of ATP [84]. Knockdown of *SLC17A9* was found to suppress the production of IL6 in THP-1 cells even after LPS stimulation suggesting an amelioration of the pro-inflammatory phenotype [85]. Notably, SLC17A9 has been found to be associated with bone marrow monopoiesis [86]. We also identified DMGs that were functionally involved in alcohol reduction (*ZADH2*), DNA-binding (*ZNF572* and *ZNF154*), RNA-processing (*SLFN13*), cytoskeletal reorganization (*TRIOBP*), brain development (*NNAT*), and keratinocytes (*KRTCAP3*), whose relation with CD is not immediately evident.

By comparing our observations with gene expression data generated by Monaco et al. [55], we found that several DMGs were differentially expressed among the three monocyte subsets, suggesting that the observed difference in methylation might have been a reflection of a difference in monocyte populations. However, the correlation between gene expression and promoter methylation is not unequivocally true, nor is the effect size of the correlation known. A more direct approach would be to compare the DNA methylome of the DMGs between non-CD, CD-active and CD-remissive for the three monocyte populations separately. While the dataset GSE73788 [87] does contain such methylation profiles, we found the results incompatible due to the availability of only a single profile per monocyte subtype, coupled with the different DNA methylation platform used.

Taken together, our observations provide interesting but preliminary insights into the manifestations of CD in the DNA methylome of circulating monocytes. We acknowledge the limited sample size of the current study. Future confirmatory studies, through for example targeted bisulfite sequencing, are necessary to validate the observations made. Additionally, mechanistic studies are required to investigate whether the differences in methylation are correlated with differences in expression, as well as whether CD activity is associated with differences in methylation of the separate monocyte subsets.

## 5. Conclusions

We have provided evidence that the DNA methylome of CD14^+^ monocytes are different between non-CD patients and CD patients, as well as between CD patients with active disease and those in remission. While the differences in DNA methylation among CD activity states are minute and the current sample size is too small to properly identify DMPs and DMRs, we observed concordant differences in methylation particular gene promoters. Future studies on the DNA methylome in circulating monocytes would have to take this into consideration when estimating the sample size necessary for a properly powered study. Our observations can to that end serve as a stepping stone in subsequent research on monocyte characteristics in CD.

## Figures and Tables

**Figure 1 jcm-09-01055-f001:**
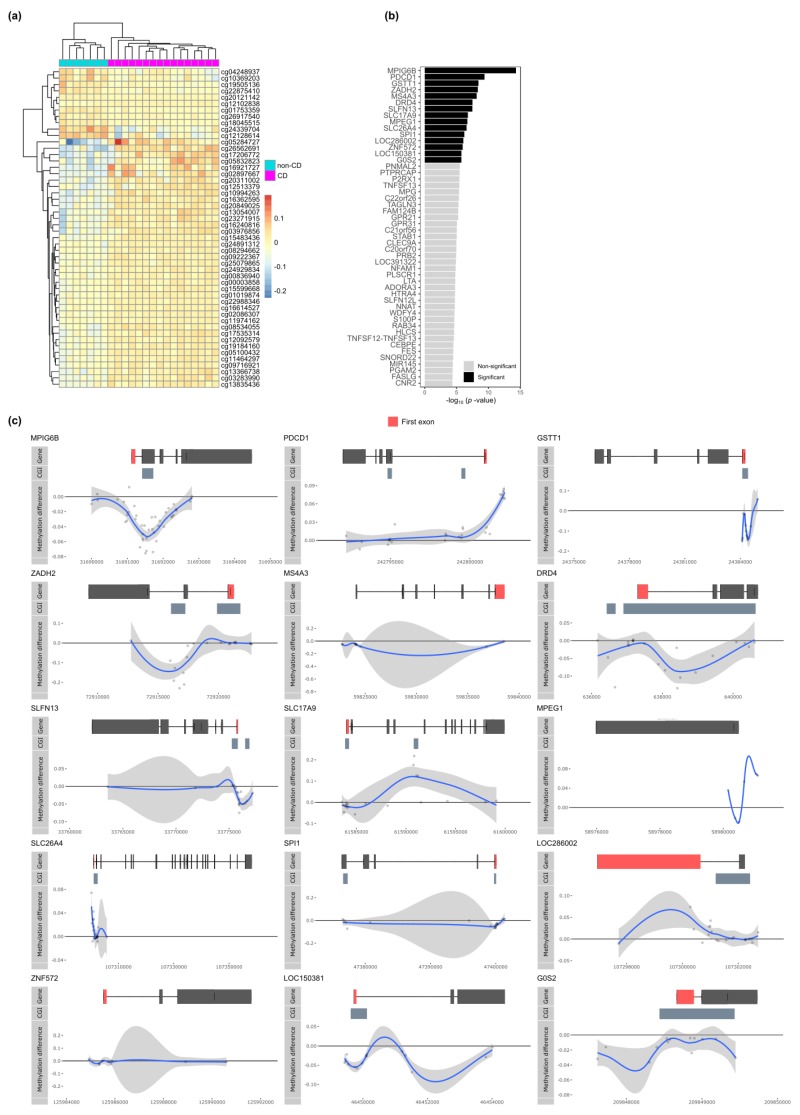
Comparing Crohn’s disease (CD) (*n* = 7) with non-CD (*n* = 16). (**a**) Heatmap organized by hierarchical clustering of the 50 most differentially methylated probes (DMPs) annotated with Illumina probe IDs. (**b**) Barplot depicting the –log_10_(*p*-value) obtained from Brown’s method for the differentially methylated genes (DMGs). Significant DMGs are indicated in black, while non-significant genes are indicated in grey. (**c**) Visualization of the significant DMGs by plotting the difference in percentage methylation relative to the position on the chromosome and the gene (“Gene”) and CpG island (“CGI”) features as obtained from UCSC. Dots represent probes on the Illumina HumanMethylation 450k BeadChip array. The blue trend line represents the loess-smoothed average across all methylation probes for the indicated region with surrounding grey area representing the standard error.

**Figure 2 jcm-09-01055-f002:**
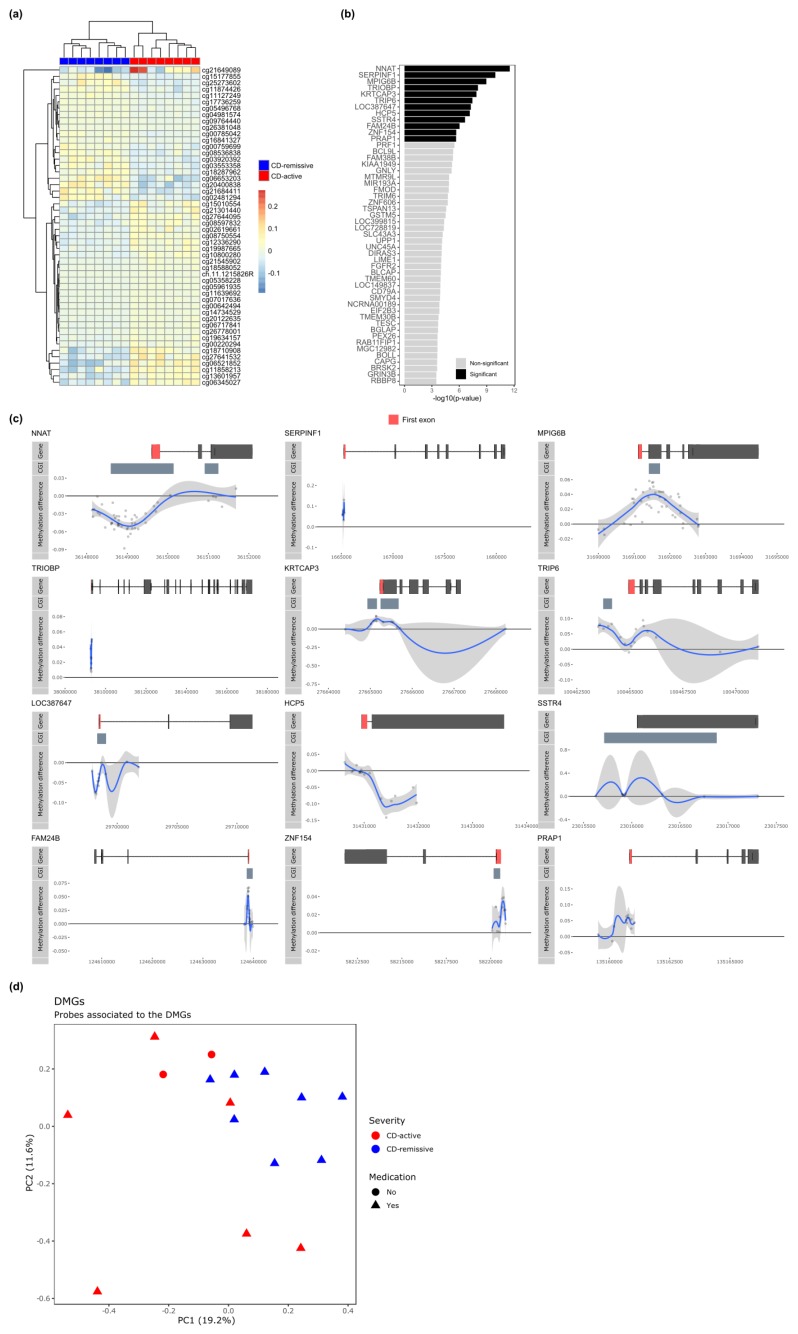
Comparing CD-active (n = 8) with CD-remissive (n = 8). (**a**) Heatmap organized by hierarchical clustering of the 50 most DMPs annotated with Illumina probe IDs. (**b**) Barplot depicting the −log_10_(*p*-value) obtained from Brown’s method for the DMGs. Significant DMGs are indicated in black, while non-significant genes are indicated in grey. (**c**) Visualization of the significant DMGs by plotting the difference in percentage methylation relative to the position on the chromosome and the gene (“Gene”) and CpG island (“CGI”) features as obtained from UCSC. Dots represent probes on the Illumina HumanMethylation 450k BeadChip array. The blue trend line represents the loess-smoothed average across all methylation probes for the indicated region with surrounding grey area representing the standard error. (**d**) Principal component analysis performed on the probes associated to the DMGs for the CD patients only.

**Figure 3 jcm-09-01055-f003:**
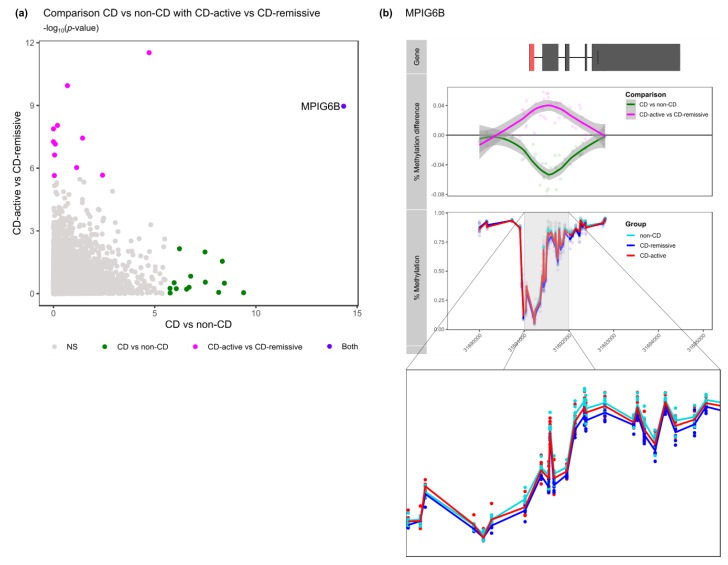
Comparison of the DMGs across the two comparisons. (**a**) Visualization of the Fisher’s combined probability test *p*-values from CD vs. non-CD on the x-axis and CD-active vs. CD-remissive on the y-axis. Colors represent the genes found to be significant in the different comparisons. (**b**) Visualization of the percentage *MPIG6B* methylation for non-CD, CD-active and CD-remissive separately with an enlarged visualization below.

**Figure 4 jcm-09-01055-f004:**
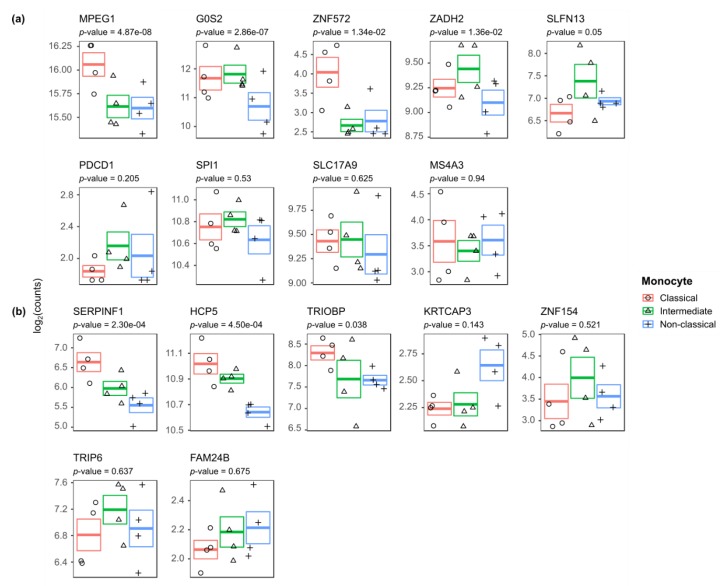
Gene expression of the (**a**) CD vs. non-CD and (**b**) CD-active vs. CD-remissive DMGs across the different monocyte subsets. Visualization of the log_2_(counts) with standard error for classical, intermediate and non-classical annotated with the *p*-value as obtained from the likelihood ratio test.

**Table 1 jcm-09-01055-t001:** Summarized patient characteristics. *p*-values were calculated through two-sided Fisher tests for binomial data and two-way ANOVAs for continuous data. Fisher tests among the medications were only performed between active and remissive CD patients. ** *p*-value < 0.01.

Characteristics				Non-CD (*n* = 7)	CD (*n* = 16)								
					Active (*n* = 8)				Remissive (*n* = 8)				*p*-Value
Female (*n*)				7	8				8				1
Age (mean years ± sd)				31.4 ± 8.34	35.7 ± 12.0				39.8 ± 4.25				0.21
CD duration (mean years ± sd)				-	9.42 ± 10.1				16.2 ± 8.54				0.169
C-reactive protein (mean mg/L ± sd)				-	22.9 ± 12.0				0.825 ± 0.79				0.00994 **
Harvey Bradshaw Index (mean ± sd)				-	6.8 ± 2.77				1.29 ± 1.8				0.00182 **
Montreal Classification (*n*)													
A1	A2			-	2	6			1	7			1
L1	L2	L3	L2+4		3	2	3	0	2	3	2	1	1
B1	B2	B3			6	2	0		6	1	1		1
P					1				3				0.5692
Any concomitant medication (*n*)				-	6				8				0.4667
Anti-TNF				-	2				6				0.1319
Corticosteroid				-	2				0				0.4667
Thiopurine				-	0				3				0.2
Questran				-	1				0				1
Celcoxib				-	1				0				1
Pantoprazole				-	1				0				1
Mercaptopurine				-	0				1				1

**Table 2 jcm-09-01055-t002:** Overview of all the DMGs found in this study alongside the relevant statistics. In short, *p*-values were obtained using Brown’s method and adjusted for multiple testing using the Benjamini–Hochberg method against all genes.

	CD vs. Non-CD		CD-Active vs. CD-Remissive	
Differentially Methylated Gene	*p*-Value	BH-Adjusted *p*-Value	*p*-Value	BH-Adjusted *p*-Value
*MPIG6B (C6orf25)*	4.63 × 10^−15^	9.19 × 10^−11^	1.08 × 10^−9^	2.15 × 10^−5^
*PDCD1*	4.05 × 10^−10^	8.04 × 10^−6^	0.905923	1
*GSTT1*	3.60 × 10^−9^	7.16 × 10^−5^	0.317294	1
*ZADH2*	4.54 × 10^−9^	9.02 × 10^−5^	0.028386	1
*MS4A3*	6.90 × 10^−9^	0.000136924	0.873469	1
*DRD4*	3.15 × 10^−8^	0.000625727	0.283934	1
*SLFN13*	3.27 × 10^−8^	0.000649811	0.010163	1
*SLC17A9*	1.67 × 10^−7^	0.003305914	0.14758	1
*MPEG1*	2.00 × 10^−7^	0.003965532	0.498594	1
*SLC26A4*	2.66 × 10^−7^	0.005275024	0.612185	1
*SPI1*	5.95 × 10^−7^	0.011817048	0.007133	1
*LOC286002*	8.54 × 10^−7^	0.016951922	0.578701	1
*ZNF572*	1.09 × 10^−6^	0.02165659	0.3022	1
*LOC150381*	1.72 × 10^−6^	0.034101222	0.935613	1
*G0S2*	1.74 × 10^−6^	0.03460972	0.560081	1
*NNAT*	1.88 × 10^−5^	0.372484	2.98 × 10^−12^	5.91 × 10^−8^
*SERPINF1*	0.204451	1	1.13 × 10^−10^	2.24 × 10^−6^
*TRIOBP*	0.634584	1	9.00 × 10^−9^	1.79 × 10^−4^
*KRTCAP3*	0.992178	1	1.31 × 10^−8^	2.61 × 10^−4^
*TRIP6*	0.036528	1	3.64 × 10^−8^	7.22 × 10^−4^
*LOC387647*	0.998103	1	5.43 × 10^−8^	1.08 × 10^−3^
*HCP5*	0.809063	1	7.04 × 10^−8^	1.40 × 10^−3^
*SSTR4*	0.873458	1	2.34 × 10^−7^	4.64 × 10^−3^
*FAM24B*	0.071767	1	9.33 × 10^−7^	0.018513
*ZNF154*	0.003772	1	2.16 × 10^−6^	0.042834
*PRAP1*	0.901922	1	2.24 × 10^−6^	0.044415

## Data Availability

The DNA methylation generated in this study has been published under controlled access for research purposes at the European Genome-phenome Archive at EGAD00010001846. All bash and R scripts have been made available on GitHub and can be found at https://github.com/ND91/PRJ0000002_CDMON. This manuscript had previously been posted as a preprint to the medRxiv preprint server at https://www.medrxiv.org/content/10.1101/2020.03.09.20033043v1.

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
