# Peer review of "Whole-Genome DNA Methylation Profiling of CD14+ Monocytes Reveals Disease Status and Activity Differences in Crohn’s Disease Patients"

_jcm, 2020, doi:10.3390/jcm9041055_

Round 1
Reviewer 1 Report
Yim et al. investigated that Identification of DNA methylation profile of CD14+ monocytes in CD patients. Even though several reports try to define the relationship between epigenetic changes and CD pathogenesis, but little has been well defined their definitive evidences in terms of epigenetic mechanism. And also, it has been questioned whether epigenetic changes are associated with clinical relevance of CD disease activity. Therefore, this manuscript addressed the important questions on how DNA methylation profile can contribute to CD patient disease activity in terms of immune cell gene expression pattern. It is well written manuscript and logically addressed what authors have found. However, there are some points authors need to think about on this project as below.
Authors have showed the data from DNA methylation profile of CD and non CD patients and compared the monocyte gene expression of candidate genes from published data. MPIG6B came up the best gene in this study which came up from genome wide DNA methylation screening platform. Authors addressed that differential methylation is mostly found in the promoter regions of candidate genes but I would recommend authors should check the CpG islands in the promoter regions of candidate genes because some genes do not have typical CpG islands in the promoter regions which means might not regulate its transcriptional gene expression in actual biology level. So that is why several technical validation experiments are necessary to prove this issue. If authors can validate the methylation level and gene expression of your candidate genes (at least MPIG6B) in CD patient tissues that you used in this study, it could make more stronger evidences of the screening tool you used here (if possible).
Minor issue
- In discussion, author can discuss the biological roles of your candidate genes in discussion section.
Author Response
We thank reviewer 1 for the time and effort in reading our manuscript and the thoughtful feedback especially given the current circumstances. Our responses can be found below. Note that the indicated line numbers are correct when “All Markup” is shown in the updated manuscript.
- “Authors addressed that differential methylation is mostly found in the promoter regions of candidate genes but I would recommend authors should check the CpG islands in the promoter regions of candidate genes because some genes do not have typical CpG islands in the promoter regions which means might not regulate its transcriptional gene expression in actual biology level.”
We acknowledge that this is an interesting point to investigate and have therefore performed the necessary analyses. We indeed found that most of the DMGs harbored CpG islands as identified by UCSC. In fact, the larger differences in methylation often occurred in the CpG islands.
We have included the CpG island annotation in the figures and results and therefore amended the methods in section 2.2 (124-126), results in section 3.2 (lines 172-175) and section 3.3 (lines 196-198), figures 1 (lines 176-186) and 2 (lines 206-217), and the discussion (lines 266-267).
- “If authors can validate the methylation level and gene expression of your candidate genes (at least MPIG6B) in CD patient tissues that you used in this study, it could make more stronger evidences of the screening tool you used here (if possible).”
We agree with the suggestion of the reviewer that a validation experiment would strengthen the observations made in the current manuscript. Unfortunately, we do not have any material leftover for a validation experiment. Instead, we would need to assemble a new cohort for validation purposes. However, we do not believe this would be feasible within the 7 days provided. As the COVID-19 outbreak has put all non-COVID-19 related lab research and patient inclusion programs on hold, we are currently not able to provide an estimate on when this would be achievable.
We have reworded our final paragraph (lines 322-330) in the discussion to emphasize that our observations, whilst novel, should be interpreted as preliminary with additional confirmatory experiments necessary.
- “In discussion, author can discuss the biological roles of your candidate genes in discussion section.”
We agree that several of the DMGs had not been adequately discussed and have therefore expanded upon this section by discussing more of the DMGs (lines 270-312). In addition, we included a paragraph on DMGs that have been described previously in a CD context (lines 271-281).
However, we would not want to embark on too much speculative thought on the downstream biological implications as we believe such statements should be supported by additional (interventional) mechanistic experiments rather than preliminary observations as presented in the current manuscript.
We have therefore reworded our vision for future research in the final paragraph of the discussion (lines 322-330), where we emphasize the need for both confirmatory studies and mechanistic studies to understand the underlying biology.
Reviewer 2 Report
I read this manuscript with interest. Authors showed tha relationship between Crohn's disease activity and DNA methylation.
- Volunteer in non-CD group has healthy status?
- Table 1. statistical comparison should be given in each parameter.
- please reveal the characteristics of CD, disease type, Montreal classification, duration etc.
- Please discuss the reproducibility of analyses.
Author Response
We thank reviewer 2 for the time and effort in reading our manuscript and providing concise critique especially given the current circumstances. Our responses can be found below. Note that the indicated line numbers are correct when “All Markup” is shown in the updated manuscript.
- “Volunteer in non-CD group has healthy status?”
That is correct, we have amended the text in the methods accordingly (line 145). We typically try to refrain from using the “healthy” moniker as volunteers were selected on the basis of not having any CD or other overt diseases. Non-overt diseases were not specifically identified prior to sampling.
- Table statistical comparison should be given in each parameter.
Table 1 (lines 151-153) has been amended accordingly where inferential statistics in the form of p-values were calculated through two-sided Fisher exact tests, in case of positive discrete data, and two-way ANOVAs, in case of continuous data.
- Please reveal the characteristics of CD, disease type, Montreal classification, duration etc.
Table 1 (lines 151-153) has been amended to include the Montreal classification with specific codes for age at diagnosis, location and the behavior of the disease. In addition, the CD duration has been provided as a separate entry as well.
- Please discuss the reproducibility of analyses.
We interpret this comment as a request on how the observations could be reproduced. We would suggest the use of targeted bisulfite sequencing to specifically investigate the regions of differential methylation as reported in the manuscript. While alternative methods such as those based on methyl-binding immunoprecipitation or third-generation sequencing would be possible as well, they currently do not provide the granularity necessary for disentangling methylation signals at the level of individual CpGs.
We acknowledge that technical validations are necessary to strengthen the reported observations. However, all material was used for the current experiment. Assembling a new cohort and performing the targeted bisulfite sequencing analyses would not be feasible within the 7 days provided. We have amended the discussion to emphasize the preliminary and observational nature of our observations and the need to validate our observations through subsequent confirmatory studies (lines 322-330).
Reviewer 3 Report
Thank you for this interesting article. You describe the rationale for your study and your methods very well. Although the overall results were not statistically significant, you recognise that the sample size was small.
I have no real improvements to suggest. I was curious about the inclusion criteria of patients with a CRP of 4 or more - this seems quite low and not particularly indicative of active disease. However, perhaps it would be helpful to include the reference range to put this in to context.
Thank you
Author Response
We thank reviewer 3 for the time and effort in reading our manuscript especially given the current circumstances. We also thank reviewer 3 for the positive comments. Our response can be found below. Note that the indicated line numbers are correct when “All Markup” is shown in the updated manuscript.
- I was curious about the inclusion criteria of patients with a CRP of 4 or more - this seems quite low and not particularly indicative of active disease. However, perhaps it would be helpful to include the reference range to put this in to context.
After rereading the manuscript and discussing the text with the team carefully, we realized that our wording and presentation of the inclusion criteria was not correct. CD activity classification was decided on clinical, endoscopic/MRI and biochemical endpoints. Most CD patients with active disease presented with a median CRP of 16. However, a single patient had a CRP level of 4.1. We classified this patient as having active CD on the basis of her MRI, which displayed an increased thickening of the mucosal wall.
We have amended the results section 3.1 (lines 147-150) accordingly where we describe the endpoints taken into consideration and have included a reference to Walsh et al. 2016 on the “Current best practice for disease activity assessment in IBD” on physician global assessment.
Round 2
Reviewer 2 Report
no comments